# Hydrological Modelling and Water Resources Assessment of Chongwe River Catchment using WEAP Model

**Tewodros M. Tena \***, **Phenny Mwaanga and Alick Nguvulu**

Department of Environmental Engineering, The Copperbelt University, Kitwe 50100, Zambia;
phenny.mwaanga@cbu.ac.zm (P.M.); anguvulu2000@yahoo.co.uk (A.N.)
*   Correspondence: tewodroslina@yahoo.com

**Abstract:** The Chongwe River Catchment (CRC) is located in Zambia. It receives a mean annual precipitation of 889 mm. The catchment is facing growing anthropogenic and socio-economic activities leading to severe water shortages in recent years, particularly from July to October. The objective of this study was to assess the available water resources by investigating the important hydrological components and estimating the catchment water balance using the Water Evaluation and Planning (WEAP) model. The average precipitation over a 52 year period and a 34 year period of streamflow measurement data for four stations were used in the hydrological balance model. The results revealed that the catchment received an estimated mean annual precipitation of 4603.12 $Mm^3$. It also released an estimated mean annual runoff and evapotranspiration of 321.94 $Mm^3$ and 4063.69 $Mm^3$, respectively. The estimated mean annual total abstractions in the catchment was 119.87 $Mm^3$. The average annual change in the catchment storage was 120.18 $Mm^3$. The study also determined an external inflow of 22.55 $Mm^3$ from the Kafue River catchment. The simulated mean monthly streamflow at the outlet of the CRC was 10.32 $m^3$/s. The estimated minimum and maximum streamflow volume of the Chongwe River was about 1.01 $Mm^3$ in September and 79.7 $Mm^3$ in February, respectively. The performance of the WEAP model simulation was assessed statistically using the coefficient of determination ($R^2 = 0.97$) and the Nash–Sutcliffe model efficiency coefficient (NSE = 0.64). The $R^2$ and NSE values indicated a satisfactory model fit and result. Meeting the water demand of the growing population and associated socio-economic development activities in the CRC is possible but requires appropriate water resource management options.

**Keywords:** hydrological components; hydrological model; water balance; streamflow; water management; Zambezi River

## 1. Introduction

### 1.1. Background

Zambia is hydrologically divided into six catchments, namely Zambezi, Kafue, Luangwa, Chambeshi, Luapula, and Tanganyika [1] as shown in Figure 1. The mean annual precipitation in Zambia ranges between 1400 mm in the north and 700 mm in the south, with an average runoff of 135 mm [2]. The north-south annual rainfall gradient is so clear that the country has been divided into three agroecological regions [3]. Region I received the lowest mean annual rainfall of less than 800 mm, Region II received intermediate mean annual rainfall between 800 and 1000 mm, while Region III received the highest annual rainfall of more than 1000 mm [4]. The estimated annual average available surface water and groundwater potential for the whole country is 237 $Mm^3$/day and 49.5 $km^3$, respectively, with little of the available surface water resources being consumed [5]. The study area,

Chongwe River catchment, is a sub catchment of the Zambezi River catchment. It falls in Regions I and II and receives a mean annual rainfall between 800 and 1000 mm [2].

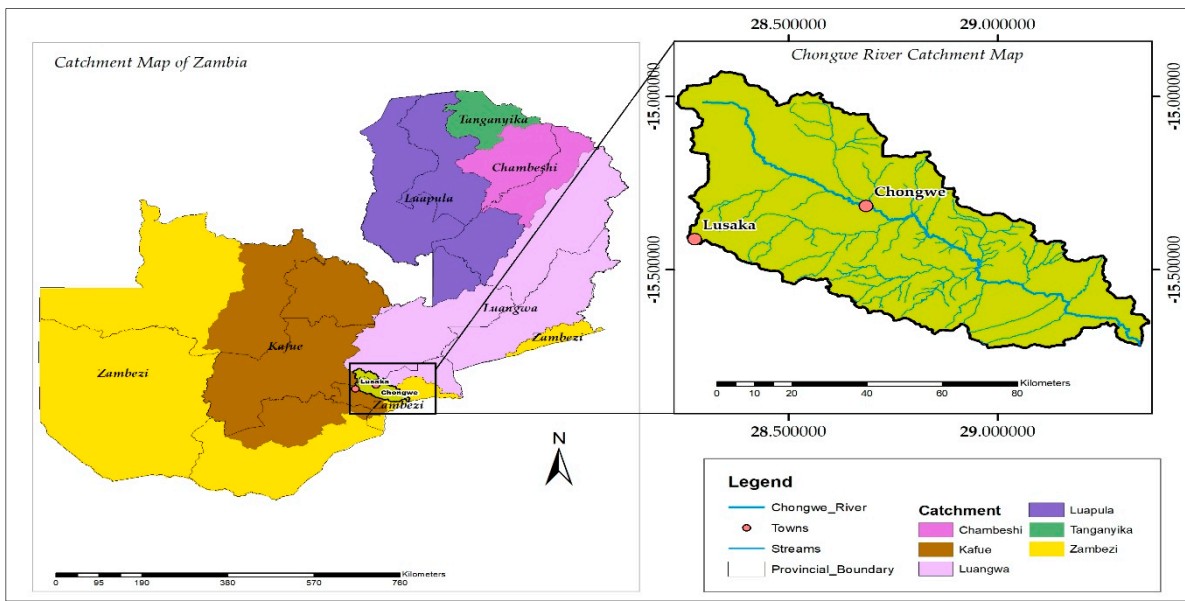

**Figure 1.** Location map of Chongwe River catchment.

The catchment, just like the whole of Zambia and Southern Africa, has faced significant changes in climatic conditions [6]. These include extreme short-term variations resulting from periodically recurring El Niño weather anomalies and long-term impacts from climate change due to global warming [7]. These changes are likely to aggravate the negative impacts on the environment and increase the stress on natural resources such as water, soils, and vegetation. The El Niño weather anomalies recur at irregular intervals of two to seven years and last from nine months to two years [8]. One of the strongest anomalies of the past 30 years, which affected the Chongwe River catchment, occurred in March 2015 and was extended through early 2016 [9]. As reported by the World Meteorological Organization (WMO), the catchment experienced a delayed start of the rainy season, irregular and below average rainfall, and above average air temperatures [10]. The subsequent effects of drought experienced were in terms of reduced agricultural productivity and water shortages from 2015 to the present [11].

There are about 834,359 people living in the Chongwe River catchment [12]. The people depend on the Chongwe River and its main tributaries for water for their domestic, agriculture, industry, and socio-economic purposes. Although the catchment receives a mean annual rainfall of 889 mm, which can be considered as abundant potential water resources, there have been severe water shortages in the last few years due to the recurrent and continuous drying up of the Chongwe River. This has in turn affected livelihoods and socio-economic activities within and surrounding communities [13].

Effective and responsible management of water resources relies on a thorough understanding of the quantity and quality of available water [14]. This study aims to provide a quantitative estimation of the available water resources, characteristics of each hydrological component, and water balance of the Chongwe River catchment with the goal of providing technical recommendations on sustainable integrated water resource management and development for the Chongwe River catchment.

*1.2. Study Area*

The Chongwe River catchment, covering an estimated area of 5168.66 km$^2$, is located between latitude 14°55′40″ to 15°43′19″ S and longitude 28°13′53″ to 29°21′24″ E as shown in Figure 1. The catchment covers parts of Lusaka, Chongwe, Chibombo, Chisamba, and Kafue Districts. It also covers 45% of Lusaka Metropolitan City. The climate of Chongwe River catchment is described as

humid subtropical, with dry winters and hot summers. The warmer wet season starts in mid-September and extends through May. Precipitation peaks high in December and January at around 232 mm/month. The colder dry season is from June through August with little or no precipitation and long dry spells. Average maximum air temperatures peak in October around 32 °C, while average minimum air temperature is 8.2 °C occurring in July. The vegetation of Chongwe River catchment is classified as Miombo woodland, dominated by semi-evergreen trees with a well-developed grass layer. The Zambezi Escarpment zone in the catchment is predominantly Mopane woodland typically interspersed by patches of Munga woodland [13]. The catchment is composed of six sub-catchments namely upper Chongwe, Ngwerere, Kanakantapa, Chalimbana, middle Chongwe–Luimba, and lower Chongwe as shown in Table 1. The main tributaries of the Chongwe River are Ngwerere, Kanakantapa, Chalimbana, and Luimba.

The Chongwe River catchment can be divided into upper, middle and lower parts. The predominant land use in the upper and middle half is agriculture and livestock production. About 6500 ha of land is now cultivated under a variety of irrigation schemes and methods in both large- and small-scale farming. The main crops grown are wheat, maize, beans, groundnut, cotton, vegetables, flowers, and horticultural crops. The other middle half is predominantly a built-up area. The lower part is mainly forest and bushland providing valuable habitat for wild animals and birds. It is also one of the ecotourism sites in Zambia. Small scale river bank cultivation and fishing are common practices by the local community in the lower part providing a means of income and household food security.

The Chongwe River catchment has potential aquifers with the most productive ones in the western and central parts [13]. Figure 2 illustrates the hydrogeological map of the catchment which shows aquifer classes and boreholes. On the basis of an aquifer classification system that was modified by Struckmeyer and Margat, the aquifer of Chongwe River catchment is categorized as a fissured aquifer and broadly belongs to three major classes namely Class C, Class D, and Class E [15]. Class C are highly productive aquifers, Class D are moderately productive aquifers, and Class E are minor aquifers with local and limited groundwater resources. These aquifers cover the larger part of the catchment area.

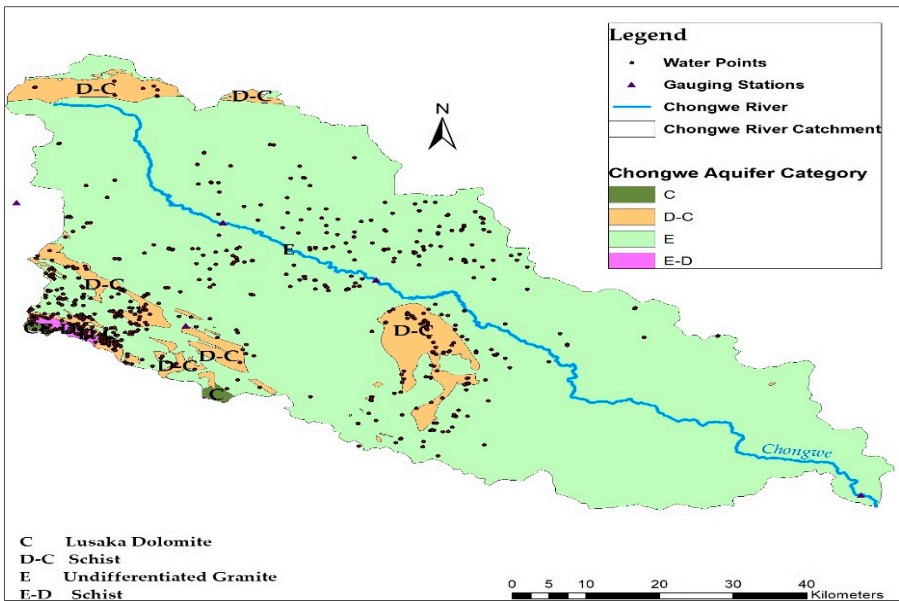

**Figure 2.** Hydrogeology map of Chongwe River catchment (modified after [13]).

**Table 1.** Sub-catchments of Chongwe River catchment.

| Name of Sub-Catchment | Area (km$^2$) |
|---|---|
| Upper Chongwe | 1236 |
| Ngwerere | 300 |
| Kanakantapa | 485 |
| Chalimbana | 674 |
| Middle Chongwe—Luimba | 1342 |
| Lower Chongwe | 1131 |

## 2. Data and Methods

### 2.1. WEAP Modelling

The Water Evaluation and Planning (WEAP) tool, developed and supported by the Stockholm Environment Institute (SEI), is a microcomputer tool for integrated water resource planning. It provides a comprehensive, flexible, and user-friendly framework for water and environment policy analysis. A growing number of water specialists and environmentalists are finding WEAP to be an important addition to their toolbox of models, databases, spreadsheets, and other software [16]. Many countries are facing challenges in the management and development of water resources. The allocation of limited water resources, water quality, catchment water protection, and appropriate policies for sustainable water use and management are the current issues concerning policy makers. The WEAP model has an integrated approach to simulate both natural and engineering components such as reservoirs, groundwater discharge, and water demand and supply. WEAP can give water planners a more comprehensive view of the broad range of factors that must be considered in managing water resources for present and future uses [17]. Operating on the basic principle of a water balance, WEAP is applicable to municipal and agricultural systems, single catchments, or complex transboundary river systems. Furthermore, WEAP can also address a wide range of issues, e.g., sectoral demand analyses, water conservation, water rights and allocation priorities, groundwater and streamflow simulations, reservoir operations, hydropower generation, pollution tracking, ecosystem requirements, vulnerability assessments, and project benefit-cost analyses [16]. WEAP aims to incorporate these values into a practical tool for water resources planning. WEAP is distinguished by its combined approach to simulating water systems and by its policy orientation. Data linked to land use, slope, soils, and vegetation are needed to simulate these hydrological processes. WEAP sets the demand and supply side of the equation in an organized way. Water use patterns, efficiencies, reuse, and allocation are placed on the demand side, while the supply side includes streamflow, groundwater, and water transfers. WEAP is a laboratory for examining alternative water development and management strategies [16]. The quantitative estimations of water availability, water demand, and water consumption both on a temporal and a spatial scale can be supported by modelling tools, capable of simulating the hydrological processes and the water management practices at catchment level for the current status, as well as for various alternative scenarios [18].

Unlike other models, WEAP offers scenario analyses in a friendly approach giving a wide range of model results in a simplified manner. The model is also a scalable tool, and it can be updated at any time. This allows for future improvement of the model results. Furthermore, WEAP is the most commonly used tool for integrated water resource management (IWRM) worldwide [16].

### 2.2. Water Balance Computation

The dynamic interaction between the various hydrological components can be examined through the investigation of hydrological processes and analysis of the catchment water balance [19]. Water balance encompasses all water inflows and outflows from a catchment area. The volume of outflows must equal the volume of inflows plus or minus storage [20]. The definition of water balance is based on the principles of the conservation of mass in a closed system or catchment.

Equation (1) was used in the computation of the annual available water balance of the catchment.

$$P + ExtIn = ET + Q + ABST \pm \Delta S \qquad (1)$$

where, P = precipitation (Mm$^3$/year), *ExtIn* = external inflow from other catchments (Mm$^3$/year), *ET* = actual evapotranspiration (Mm$^3$/year), *Q* = streamflow (Mm$^3$/year), *ABST* = abstraction (Mm$^3$/year), $\Delta S$ = change in storage (Mm$^3$/year).

The WEAP model was used in the estimation of the components of the water balance in Equation (1) using the climate, physical, and hydrologic inputs from the Chongwe River catchment.

Evapotranspiration included evaporation losses from land surfaces, vegetation, and open water bodies. Streamflow comprised both direct runoff and baseflow. The water cycle also consisted of groundwater storage reservoirs called aquifers. The recharge of these groundwater aquifers accounted for entries from rainfall (direct recharge) as well as from influent seepage from rivers [21].

*2.3. WEAP Model Data Inputs*

2.3.1. Climate and Physical Data

The climatic data inputs used in the WEAP model were precipitation, air temperature, relative humidity (RH), wind speed, and solar radiation. Solar radiation in WEAP can be quantified by entering solar radiation data or hours of sunshine per day or cloudiness fraction depending on data availability. In this study, cloudiness fraction data were used as an input. These data inputs were obtained from the Zambia Meteorological Department (ZMD), the Global Weather Net and the Southern African Science Service Centre for Climate Change and Adaptive Land Management (SASSCAL) Weather Net for a period from 1965 to 2017. The WEAP model used precipitation in the calculation of evapotranspiration, streamflow, and baseflow. The model also used average mean air temperature, relative humidity, average wind speed, and cloudiness fraction in the calculation of evapotranspiration. These data inputs are presented in Table 2.

**Table 2.** Averaged monthly climate values of Chongwe River catchment (1965 to 2017).

| Climate Variable | Month | | | | | | | | | | | |
|---|---|---|---|---|---|---|---|---|---|---|---|---|
| | Oct | Nov | Dec | Jan | Feb | Mar | Apr | May | Jun | Jul | Aug | Sep |
| Average Precipitation (mm) | 18 | 98 | 212 | 232 | 201 | 93 | 28 | 5 | 0 | 0 | 0 | 2 |
| Average Air Temperature (°C) | 17.8 | 17.7 | 16.6 | 14.6 | 11.4 | 8.7 | 8.2 | 10.3 | 14.1 | 17.4 | 18.2 | 18 |
| Average RH (%) | 39.34 | 52.4 | 73.4 | 83.71 | 85.7 | 82.45 | 80.14 | 69.17 | 63.91 | 58.05 | 48.1 | 39.77 |
| Average Wind Speed (m/s) | 39.34 | 52.4 | 73.4 | 83.71 | 85.7 | 82.45 | 80.14 | 69.17 | 63.91 | 58.05 | 48.1 | 39.77 |
| Cloudiness Fraction | 0.5 | 0.3 | 0.1 | 0.1 | 0.1 | 0.3 | 0.4 | 0.7 | 0.9 | 1 | 1 | 0.7 |

Physical data inputs were land use/land cover data and soil physical properties of the catchment. The land use/land cover data were generated from the United States Geological Survey (USGS) Landsat 8 Operation Land Imager (OLI) using an Earth Resource Data Analysis System (ERDAS Imagine) 2014 and analyzed using ArcMap 10.3 [22,23]. The maximum likelihood classification algorithm was used to classify the catchment into five land use classes as shown in Figure 3. Table 3 shows the land use/land cover class distribution for the years 1984 and 2017. Land use data were important in WEAP to imitate the hydrological relations between the soil, atmosphere, and runoff. It was vital in the algorithm for computing evapotranspiration.

The soil physical properties used in the WEAP model were soil texture, soil water holding capacities, soil horizon depth, and conductivity. These were extracted from the 1:1 Million Soil Map of Zambia [24], documented field measurements from the Zambian Agricultural Research Institute, and the researchers own field survey measurements.

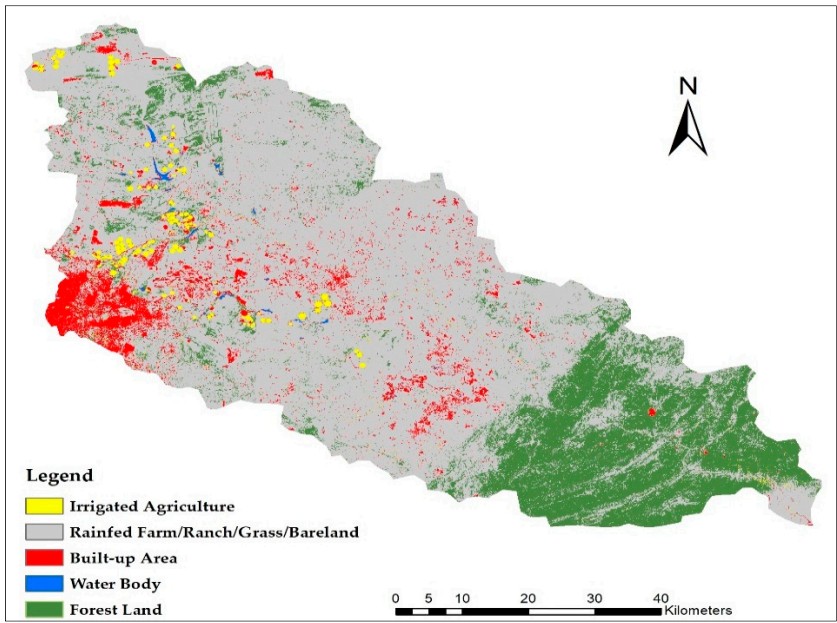

**Figure 3.** Land use/land cover map of Chongwe River catchment.

**Table 3.** Land use/land cover classification of Chongwe River catchment.

|  | 1984 | | 2017 | |
| --- | --- | --- | --- | --- |
| **Land Use/Land Cover Class** | **Area (km$^2$)** | **%** | **Area (km$^2$)** | **%** |
| Irrigated agriculture | 7.54 | 0.15 | 63.76 | 1.23 |
| Rainfed farm/ranch/grass/bare land | 3269.86 | 63.26 | 3749.41 | 72.55 |
| Built-up area | 60.11 | 1.16 | 289.95 | 5.60 |
| Forest land | 1792.15 | 34.67 | 1055.44 | 20.42 |
| Water body | 39.04 | 0.76 | 10.10 | 0.19 |
| Total | 5168.66 | | 5168.66 | |

### 2.3.2. Hydrologic Data

The hydrologic parameters used were streamflow, surface water abstraction, groundwater abstraction, external inflow data, and evapotranspiration. Evapotranspiration data were derived from climatic and land use data using the Penman–Monteith equation in WEAP.

Streamflow Data for Great East Bridge on Chongwe River were collected from the Zambia Water Resources Management Authority (WARMA). The data were used for calibration of the WEAP model. Periodical streamflow measurement at the Great East Bridge and at the outlet of the tributaries were conducted and the discharge volumes were estimated. Good estimations of parameters and initial state variables were essential to enable the hydrological models to make accurate estimations [25].

The surface water abstractions data were grouped into abstractions for irrigation, livestock, and industrial use. The data for irrigation were obtained from WARMA water permit database, satellite imagery, and field assessments, while the data for livestock and domestic/industrial use were obtained from the Ministry of Agriculture and the Lusaka Water and Sewerage Company (LWSC), respectively.

Groundwater abstraction data for rural water supply, industry, domestic and irrigation purposes were collected from WARMA and the Groundwater Resources Management Support Programme (GReSP)/Bundesanstalt für Geowissenschaften und Rohstoffe (BGR) database and analyzed using GeODin software [26] and Microsoft excel. Rural water demand encompassed all domestic-type water requirements outside the urban areas [27]. In WEAP, the monthly variation of water abstraction for irrigation was derived as a function of rainfall as illustrated in Figure 4. Supplementary irrigation was also a common practice during shortage of rainfall.

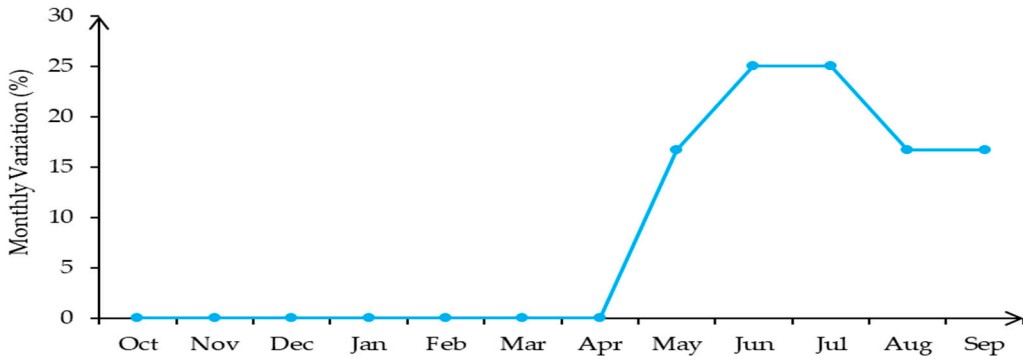

**Figure 4.** Monthly variation of irrigation water abstraction for Chongwe River catchment.

External inflow to the Chongwe River catchment, mainly from the Kafue River catchment, was estimated using wastewater discharge information obtained from LWSC and described in Table 4. This was incorporated in the WEAP through the inflow option for wastewater. The mean discharge of wastewater into the Chongwe River catchment through Ngwerere Stream is 0.9533 m³/s. According to LWSC, 77.4% of the total water supply to Lusaka City was from the Kafue River catchment and 22.6% was from the Chongwe River catchment. Therefore, about 77.4% of the wastewater discharged into the Chongwe River via the Ngwerere Stream originated from the Kafue River catchment.

**Table 4.** Wastewater discharge from the treatment plants into Ngwerere Stream (2011–2017).

| Month | Mean Discharge Flow (m³/s) at Each Treatment Plant | | | | |
|---|---|---|---|---|---|
| | Machinchi | Ngwerere Ponds | Chelstone Ponds | Kaunda Square Ponds | Total |
| Jan | 0.9469 | 0.1576 | 0.0185 | 0.0247 | 1.1477 |
| Feb | 0.8777 | 0.2075 | 0.0191 | 0.0255 | 1.1298 |
| Mar | 0.8625 | 0.1296 | 0.0133 | 0.0177 | 1.0231 |
| Apr | 0.8106 | 0.1545 | 0.0123 | 0.0164 | 0.9938 |
| May | 0.8130 | 0.0957 | 0.0093 | 0.0125 | 0.9305 |
| Jun | 0.7416 | 0.0809 | 0.0123 | 0.0164 | 0.8512 |
| Jul | 0.6783 | 0.2411 | 0.0078 | 0.0103 | 0.9375 |
| Aug | 0.7082 | 0.0984 | 0.0088 | 0.0117 | 0.8271 |
| Sep | 0.7191 | 0.1035 | 0.0082 | 0.0110 | 0.8418 |
| Oct | 0.6299 | 0.0816 | 0.0065 | 0.0086 | 0.7266 |
| Nov | 0.7699 | 0.0835 | 0.0071 | 0.0095 | 0.8700 |
| Dec | 0.7987 | 0.3045 | 0.0240 | 0.0320 | 1.1592 |
| Mean flow (m³/s) | 0.7797 | 0.1449 | 0.0123 | 0.0164 | 0.9533 |

*2.4. Schematization of Chongwe River Catchment in the WEAP Model*

The Chongwe River catchment boundary was delineated using WEAP catchment delineation mode and the generated schematic map further refined from a GIS based vector map of the Chongwe River catchment area and river network. To model the Chongwe River catchment in the WEAP, further datasets consisting of tributary streamflow data and groundwater data analyzed from 802 boreholes using GeODin software were obtained. Six groundwater demand nodes were created in the WEAP for the demand analysis of the Chongwe River catchment and computation of the recharge, abstraction rate, and volume. Note that each demand node represented a specific group of water users. The six demand nodes in this study represented (1) irrigation demand for upper and middle part, (2) irrigation demand for downstream, (3) domestic demand for urban and rural water use, (4) water demand for Chongwe Town water supply, (5) livestock water demand, and (6) irrigation water demand from groundwater. As shown in Figure 5, all the six water demand nodes depended on water from the catchment surface and groundwater resources. The time step for the WEAP model was based on calendar month with the

hydrological year starting in the month of October and ending in September (Figure S1, Supplementary Materials).

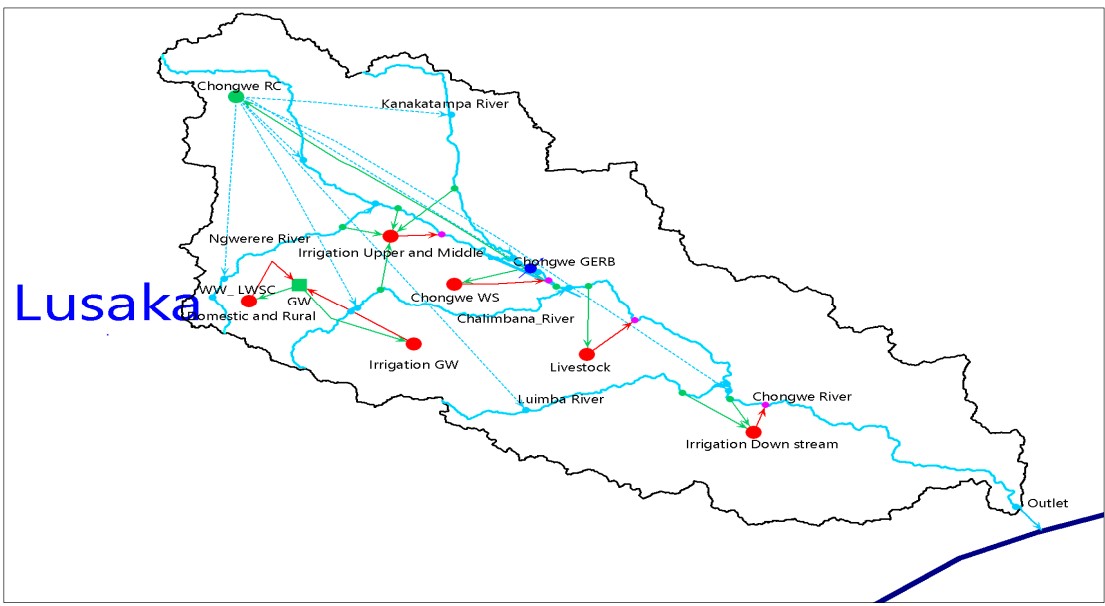

**Figure 5.** WEAP schematic map for Chongwe River catchment.

*2.5. WEAP Model Performance*

The WEAP model was calibrated using the observed streamflow data for the Chongwe Great East Road Bridge gauging station obtained from WARMA for a period from the 1982/83 to 2016/17 hydrological year. The comparison was made between the observed and simulated streamflow data for the assessment of the accuracy of the model by calculating the coefficient of determination ($R^2$) and the Nash–Sutcliffe model efficiency coefficient (NSE). $R^2$ was determined using the statistical formulas in Microsoft excel while NSE was computed using Equation (2).

$$\text{NSE} = 1 - \frac{\sum_{t=1}^{T}\left(Q_s{}^t - Q_0{}^t\right)^2}{\sum_{t=1}^{T}\left(Q_0{}^t - \overline{Q}_0\right)^2} \tag{2}$$

where, $\overline{Q}_0$ is the mean of the observed streamflow, $Q_s{}^t$ and $Q_0{}^t$ are simulated and observed streamflow at time t respectively.

*2.6. Field Survey and Observations*

Quarterly based field assessments and survey were conducted. These involved participatory field observation on various features of the catchment, streamflow measurement, water sampling, soil survey, data collection from gauging stations, assessment of water abstraction rate, field farm survey, and focus group discussion with the community members of the Chongwe River catchment.

## 3. Results and Discussion

*3.1. Results*

### 3.1.1. Evapotranspiration

Long term actual and potential evapotranspiration monthly values as obtained from the WEAP model are presented in Table 5. The Chongwe River catchment was estimated to have an annual average actual and potential evapotranspiration of 4063.68 Mm$^3$ (786 mm) and 6061.88 Mm$^3$ (1172.81 mm) per

year, respectively. In addition, the results showed that potential evapotranspiration (PET) throughout the year was higher than actual evapotranspiration (AET). This was also detectable from the equations in Figure 6, where actual evapotranspiration has been decreasing at a rate of 0.24 Mm$^3$/annum while potential evapotranspiration has been increasing at a rate of 0.03 Mm$^3$/annum.

**Table 5.** WEAP simulated mean monthly potential and actual evapotranspiration for Chongwe River catchment.

| Months | Potential ET | | Actual ET | |
|---|---|---|---|---|
| | Mm$^3$ | mm | Mm$^3$ | mm |
| Oct | 697.00 | 134.85 | 270.90 | 52.41 |
| Nov | 622.03 | 120.35 | 254.53 | 49.25 |
| Dec | 509.60 | 98.59 | 302.45 | 58.52 |
| Jan | 445.08 | 86.11 | 358.70 | 69.40 |
| Feb | 401.08 | 77.60 | 372.77 | 72.12 |
| Mar | 532.49 | 103.02 | 512.48 | 99.15 |
| Apr | 481.52 | 93.16 | 445.99 | 86.29 |
| May | 483.55 | 93.55 | 411.60 | 79.63 |
| Jun | 365.56 | 70.73 | 277.95 | 53.78 |
| Jul | 419.12 | 81.09 | 281.46 | 54.46 |
| Aug | 525.70 | 101.71 | 301.40 | 58.31 |
| Sep | 579.14 | 112.05 | 273.45 | 52.91 |
| SUM | 6061.88 | 1172.81 | 4063.68 | 786.22 |

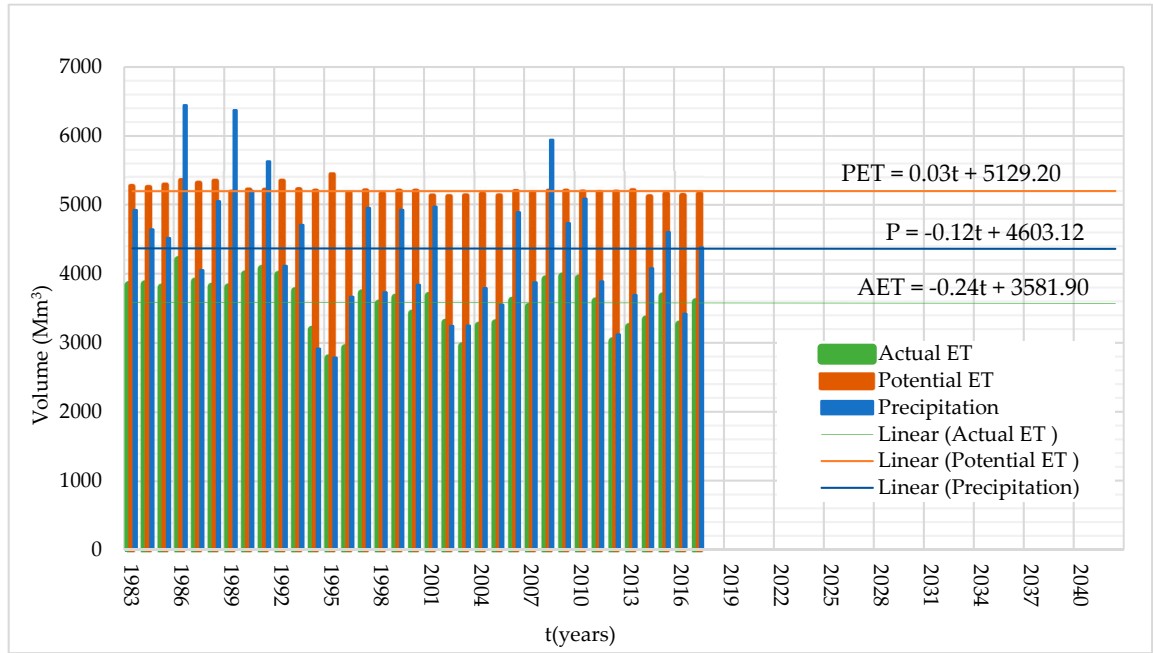

**Figure 6.** Annual actual evapotranspiration (ET), potential ET and precipitation for Chongwe River catchment.

## 3.1.2. Streamflow and Baseflow at the Outlet

During the period under study, the WEAP Model simulated average streamflow was 10.32 m$^3$/s. The minimum streamflow volume was 1.01 Mm$^3$ in September and the maximum streamflow volume was 79.68 Mm$^3$ in February. The trend indicates that on average annual streamflow at the outlet has increased at a rate of 0.13 Mm$^3$ per annum (Figure 7).

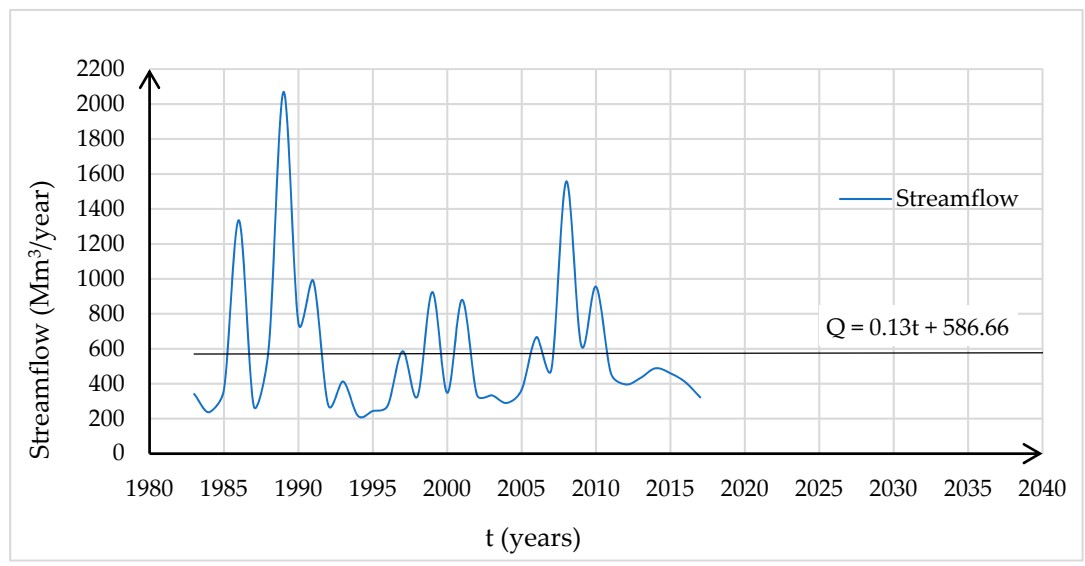

**Figure 7.** Annual streamflow variation at the outlet of Chongwe River Catchment.

The WEAP Model simulated average annual baseflow at the outlet of the Chongwe River catchment was 205.57 $Mm^3$. The minimum baseflow volume was 0.77 $Mm^3$ in October and the maximum baseflow volume was 54.76 $Mm^3$ in March. Figure 8 presents the monthly streamflow and baseflow of the Chongwe River catchment during the period under study.

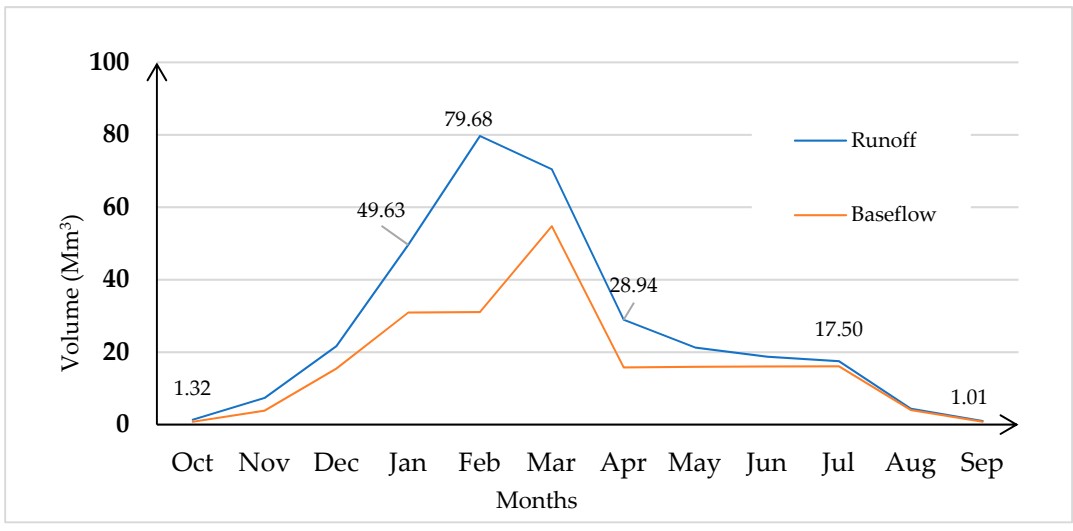

**Figure 8.** Monthly runoff and baseflow in Chongwe River Catchment.

3.1.3. Water Abstractions

The analyzed surface water and groundwater abstractions from all the demand nodes (water users) in the catchment are presented in Figure 9 which shows that the total volume of water abstracted for the hydrological year 2016/17 was 119.87 $Mm^3$.

Surface water abstraction includes irrigation, Chongwe Town water supply, livestock, and maintenance for ecosystem demand. The components of surface water use and respective volumes of abstraction are given in Table 6. The total volume of surface water abstraction was 90.21 $Mm^3$/year.

The abstraction amounts for groundwater use components of irrigation, domestic, and rural water supply are given in Table 7. The total volume of groundwater abstraction was 29.67 $Mm^3$/year.

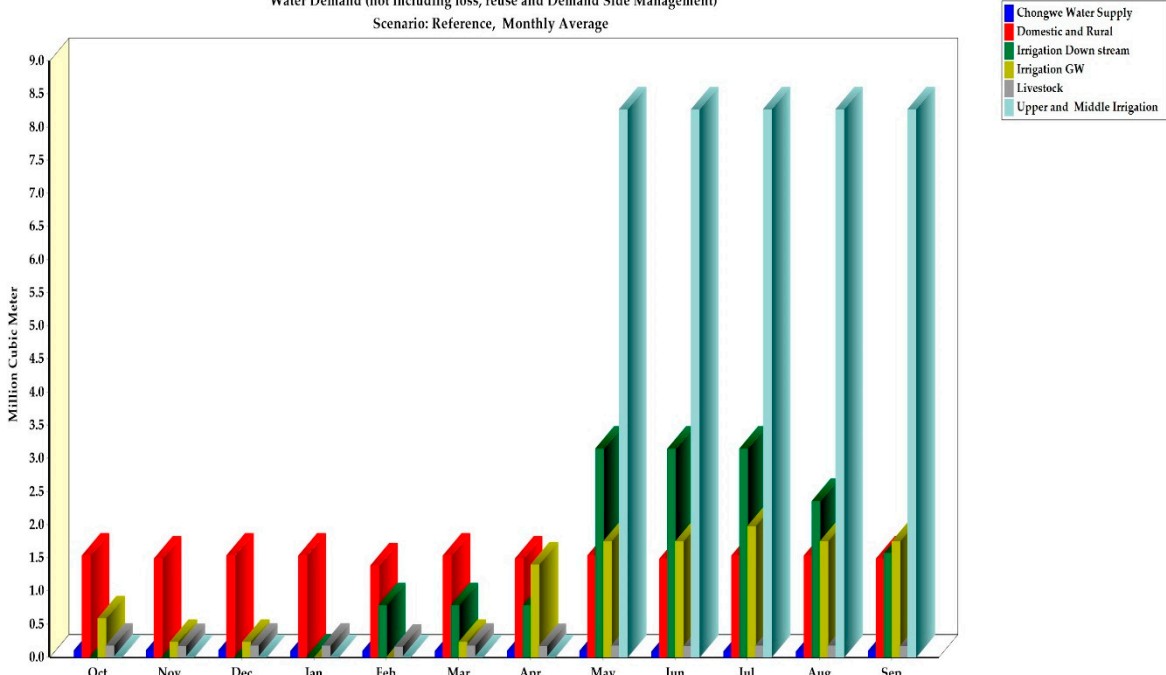

**Figure 9.** Water abstractions at the demand nodes in the WEAP model.

**Table 6.** Total volume of surface water abstraction for Chongwe River Catchment (obtained from the WEAP).

| Months | Water Supply (Mm$^3$) | Downstream Irrigation (Mm$^3$) | Livestock Water Use (Mm$^3$) | Upper and Middle Catchment Areas Irrigation (Mm$^3$) | Eco System Maintenance (Mm$^3$) | Total Surface Water Abstraction Volume (Mm$^3$) |
|---|---|---|---|---|---|---|
| Oct | 0.10 | 0.00 | 1.71 | 0.00 | 7.51 | 7.78 |
| Nov | 0.10 | 0.00 | 0.17 | 0.00 | 2.40 | 2.67 |
| Dec | 0.10 | 0.00 | 0.17 | 0.00 | 0.00 | 0.27 |
| Jan | 0.01 | 0.00 | 0.17 | 0.00 | 0.00 | 0.27 |
| Feb | 0.10 | 0.79 | 0.15 | 0.00 | 0.00 | 1.04 |
| Mar | 0.10 | 0.79 | 0.17 | 0.00 | 0.00 | 1.05 |
| Apr | 0.10 | 0.79 | 0.17 | 0.00 | 0.00 | 1.05 |
| May | 0.09 | 3.14 | 0.17 | 8.26 | 2.70 | 14.37 |
| Jun | 0.09 | 3.14 | 0.17 | 8.26 | 3.30 | 14.96 |
| Jul | 0.09 | 3.14 | 0.17 | 8.26 | 3.60 | 15.27 |
| Aug | 0.09 | 2.36 | 0.17 | 8.26 | 4.50 | 15.38 |
| Sep | 0.09 | 1.57 | 0.17 | 8.26 | 6.00 | 16.10 |
| Total abstraction volume (Mm$^3$) | 1.14 | 15.71 | 2.01 | 41.31 | 30.02 | 90.21 |

**Table 7.** Total volume groundwater abstraction for Chongwe River Catchment (obtained from the WEAP).

| Months | Groundwater Abstraction Irrigation (Mm$^3$) | Groundwater Abstraction Domestic and Rural Water Supply (Mm$^3$) | Total Groundwater Abstraction Volume (Mm$^3$) |
|---|---|---|---|
| Oct | 0.58 | 1.53 | 2.11 |
| Nov | 0.23 | 1.48 | 1.72 |
| Dec | 0.23 | 1.53 | 1.76 |
| Jan | 0.00 | 1.53 | 1.53 |
| Feb | 0.00 | 1.38 | 1.38 |
| Mar | 0.23 | 1.53 | 1.76 |
| Apr | 0.14 | 1.48 | 2.88 |
| May | 0.17 | 1.53 | 3.28 |
| Jun | 0.17 | 1.48 | 3.23 |
| Jul | 0.20 | 1.53 | 3.51 |
| Aug | 0.17 | 1.53 | 3.28 |
| Sep | 0.17 | 1.48 | 3.23 |
| Total abstraction volume (Mm$^3$) | 11.63 | 18.04 | 29.67 |

### 3.1.4. Summary of the Hydrological Water Balance

The hydrological water balance for Chongwe River Catchment developed from the WEAP model is as presented in Figure 10 for the year 2016/17 and Table 8 for selected years.

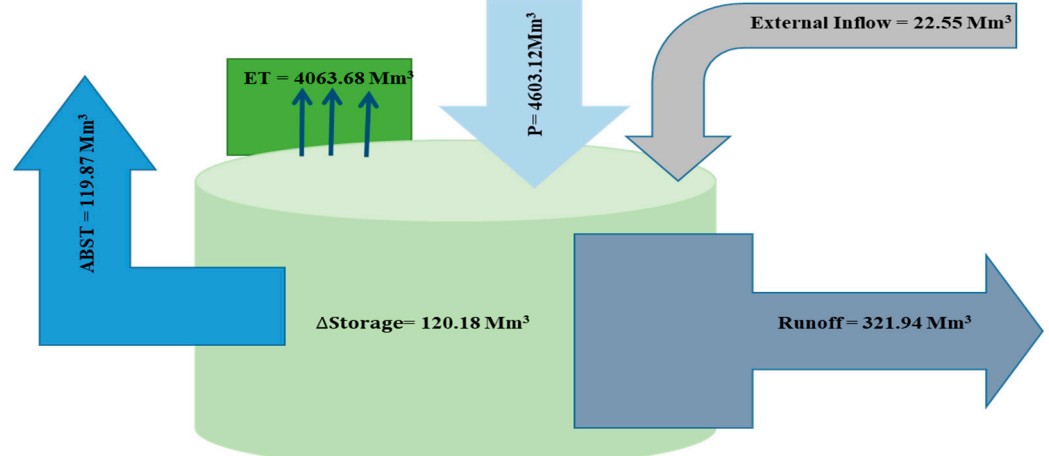

**Figure 10.** Water balance model for Chongwe River Catchment for 2016/17.

**Table 8.** Estimated annual water balance components for Chongwe River Catchment for selected years.

| Year | 1983/84 | 1993/94 | 2003/04 | 2014/15 | 2016/17 |
|---|---|---|---|---|---|
| Precipitation (Mm$^3$) | 4638.38 | 2982.02 | 4594.85 | 4611.22 | 4603.13 |
| External inflow (Mm$^3$) | * | * | * | 22.55 | 22.55 |
| Evapotranspiration (Mm$^3$) | 4344.83 | 3090.98 | 4098.16 | 4068.17 | 4063.68 |
| Streamflow (Mm$^3$) | 236.59 | 216.59 | 290.01 | 459.70 | 321.94 |
| Abstractions (Mm$^3$) | 46.09 | 56.61 | 79.32 | 112.36 | 119.87 |
| Change in storage (Mm$^3$) | 10.87 | −382.16 | 127.36 | −6.46 | 120.18 |

\* No data available.

### 3.1.5. Model Performance

The graphical comparison of the monthly average observed streamflow with the simulated streamflow for a period from 1982/83 to 2016/17 are presented in Figure 11. The model fit was assessed using the coefficient of determination ($R^2$) (Figure 12) and Nash–Sutcliffe model efficiency coefficient (NSE). From the comparison, an $R^2$ of 0.97 and NSE of 0.64 were achieved. Computation of NSE is shown in Table S1 under the Supplementary Materials.

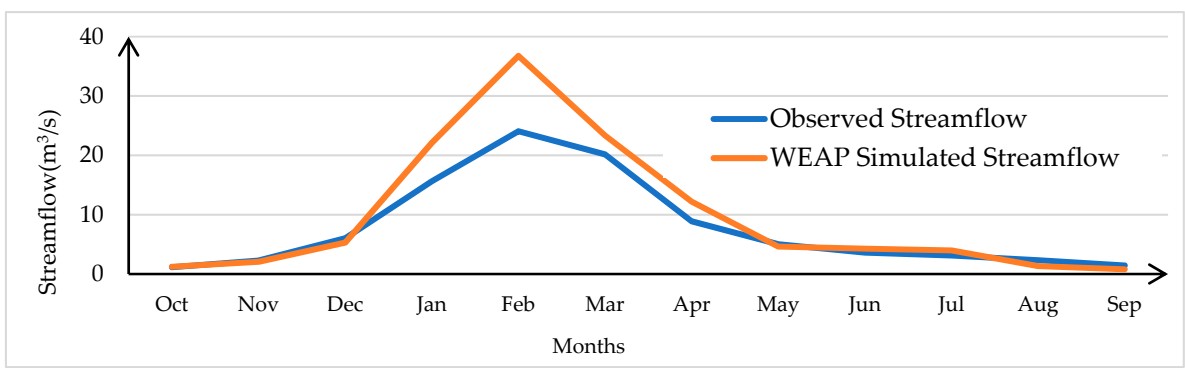

**Figure 11.** Comparison of observed and simulated mean monthly streamflow at Chongwe Great East Road Bridge (Station 5-025) for the period 1982/83 to 2016/17.

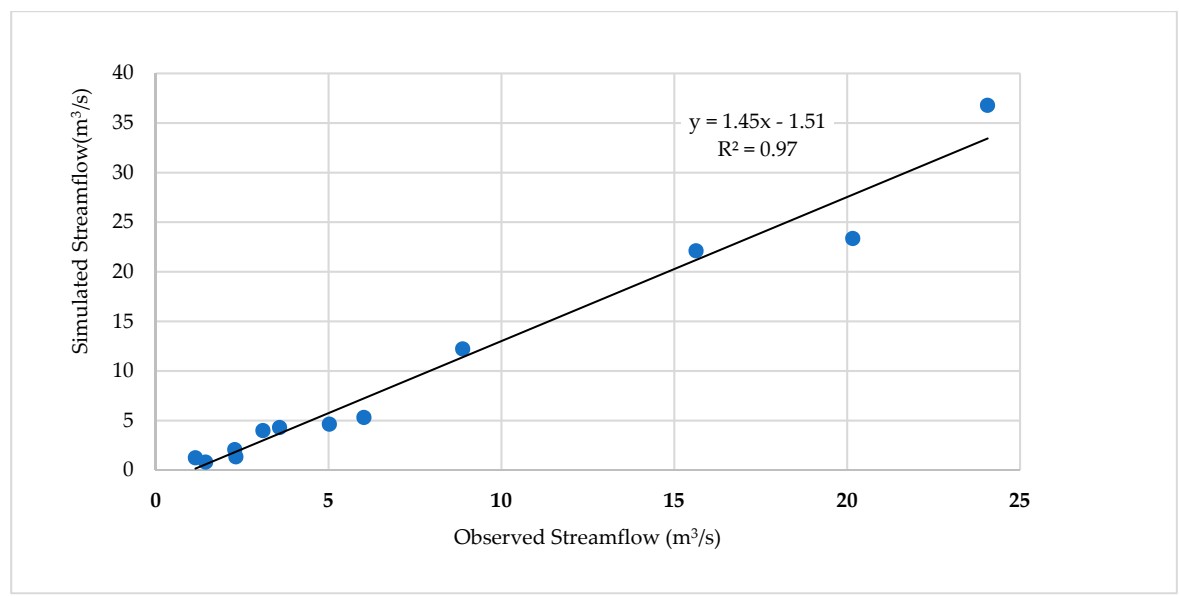

**Figure 12.** Comparison of observed and simulated streamflow using $R^2$.

*3.2. Discussion*

Hydrological Water Balance for Chongwe River Catchment and Its Components

The model shows that the major component of water inflow into the catchment is precipitation (Figure 10). The observed trend for a period of 34 years shows that precipitation is decreasing at a rate of 0.12 Mm³/year. This decrease agrees with the findings in the study by Chisola et al. [28] in which the decrease was attributed to climate change.

The model further shows another external water inflow consisting of a considerable volume of water transferred from the Kafue River catchment into Chongwe River catchment through the LWSC System. The Lusaka Water Sewerage Company abstracts 226,000 m³ of water per day of which 96,000 m³ is from the Kafue River and 130,000 m³ is from groundwater sources [29]. The production boreholes located in Chongwe River catchment supplies about 51,024 cubic meters of water per day of water which is 22.6% of the total water supply of Lusaka City [30]. Therefore, the volume of water transferred from the Kafue River catchment into the Chongwe River catchment through LWSC system is about 77.4% of the total wastewater discharge. The wastewater is discharged from four treatment plants of Lusaka City into the Ngwerere Stream, which is one of the tributaries of the Chongwe River. Table 4 shows that a total volume of 0.9533 m³/s of wastewater is discharged into the catchment. According to Figure 10, the external inflow contributes about 18.8% of the total abstractions in the catchment. This indicates that the external inflow plays an important role in the hydrological balance of the catchment especially in the dry periods.

As shown in the water balance model, the largest component of water outflow is evapotranspiration amounting to 786 mm (88% of precipitation). Evapotranspiration is too high compared to the limited volume of precipitation. This agrees with the findings in the National Water Resources Master plan of Zambia which indicates that the average actual and potential evapotranspiration for Lusaka region are 739 mm and 1394 mm, respectively [5]. The high evapotranspiration rate can be partly attributed to irrigation farming during the dry season due to the expansion of center-pivot, overhead sprinkler, and basin methods of irrigation systems adopted by both commercial and small-scale farmers in the Chongwe River catchment [31]. These systems of irrigation expose water to the atmosphere and under action of wind and high air temperatures, the water is evaporated thereby increasing the rate of evapotranspiration. According to Nick [13], there are several small dams along the Chongwe River and its tributaries constructed for the purpose of irrigation. The assessment indicated that there was high competition for water in upper and middle part of the catchment from May to September, due to

an increased water demand for irrigation (Figure 9). This has resulted in less water and low streamflow of the Chongwe River on the upper and middle part of the catchment.

The other outflow component of significant effect in the water balance model is the streamflow at the outlet of the catchment. This is the amount of water which leaves the catchment as surface runoff and baseflow. Comparing averaged monthly values of streamflow with precipitation, it is clear that streamflow variation responds to the pattern of precipitation. As precipitation increases, streamflow increases, and vice versa. However, in the dry season the baseflow is the major contributor of streamflow (Figure 8). This is in line with the result that showed there is high runoff in rainy seasons and very low in dry period. The study conducted by Chisola et al. [28] on the upper catchment of the Chongwe River also indicated that there is an increase in runoff during wet season flows and a reduction in dry season flows. An analysis of the long-term time series variation of streamflow at the outlet of the catchment for a period of 34 years (Figure 7) indicates that the streamflow is increasing at a rate of about 0.13 $Mm^3$ per annum. This increase can be attributed to the changes in land use/land cover as shown in Table 3. The built-up area increased from 1.16% in 1984 to 5.60% in 2017. Construction of buildings and road infrastructure increases the runoff coefficient, thereby increasing the rate of streamflow. Streamflow data are most important for the execution of water resource plans and models [32].

In addition to evapotranspiration and streamflow, the total water abstraction was considered an outflow component of the water balance. The total water abstraction amount was 119.87 $Mm^3$ per year. This is composed of both groundwater and surface water abstractions for the purpose of irrigation, domestic, rural, and livestock uses. Of these, the largest water abstractions are mainly for irrigation purposes [33]. It is expected that the quantity of water abstraction will increase due to an increase in irrigation area.

The annual change in storage for the catchment was varying during the study period (Table 8). There was a positive change in storage in the years 1983/84, 2003/04 and 2016/17. The positive change in storage was a result of high available precipitation despite high evapotranspiration. However, in the years 1993/94 and 2014/15, there was a negative change in storage which was due to little precipitation. These indicate that the change in storage responds largely to temporal variation in precipitation. In the hydrological year 2016/17, the change in storage was about 2.6% of the total water inflow which is expected to decrease with the decreasing precipitation trend (Figure 6) and with the development of socio-economic activities.

## 4. Conclusion and Recommendations

### 4.1. Conclusions

The hydrological components for the Chongwe River catchment were analyzed and the water balance was determined using the WEAP model. The available water resources of the Chongwe River catchment were also estimated. The performance of the WEAP model simulation was assessed statistically through the computation of the coefficient of determination ($R^2$) and the Nash–Sutcliffe model efficiency coefficient (NSE). An $R^2$ of 0.97 and NSE of 0.64 were achieved indicating a satisfactory model fit. The model established that the Chongwe River catchment receives precipitation plus an external inflow of 22.55 $Mm^3$/year from the Kafue River catchment. About 88% of water leaves the catchment through evapotranspiration and 6.7% as streamflow. Only 2.6% of water is abstracted from the available water resources of the catchment for various uses. The total abstraction of all water uses in the catchment can also be expressed as 37.2% of the total runoff. The hydrological assessment also indicated that there is less water on the upper and middle part of the catchment where there is high competition for water in the dry season due to an increase in commercial and small-scale irrigation farms. Even though the streamflow increases during the rainy season, most farmers do not have enough reservoirs to harvest the runoff for use during the dry season. On the basis of our results and observations, meeting the water demand of the growing population and associated

socio-economic development activities in the catchment is possible but requires appropriate water resources management interventions.

*4.2. Recommendations*

The result of this study revealed that runoff at the outlet of the catchment has been increasing. The increase peaks during the wet season and subsides during the dry season. Furthermore, the study indicated that wastewater discharged into the Ngwerere stream plays a considerable role on the water balance of the catchment. Therefore, it is advisable to regularly monitor the water quality of effluents before discharging into the stream based on the environmental management standards. The study also revealed a drastic change of forest land to built-up area and farm land (Table 3). It is important to protect and increase the forest reserve in order to ensure sustainable recharge of the head water. The peak runoff during the rainy period is directly influenced by the change of land use and deforestation.

With intensified irrigation activities in the upstream and middle parts of the Chongwe River catchment during the dry period there is need for appropriate water management options for sustainability of the ecosystem of the river. The options suggested for the Chongwe River catchment include (i) implementation of water harvesting technologies such as micro dams, ponds, weirs, and check dams to harvest excess runoff in the wet season to help in overcoming the water deficit during the dry season, (ii) introduction of groundwater recharge ponds and protection of recharge areas for sustainability of groundwater resources and baseflow of Chongwe River, (iii) development and implementation of integrated catchment management strategies in collaboration with all water users and the community, and (iv) institutionalization and establishment of the Catchment Water Users Association and the River Management Council through the Water Resources Management Authority (WARMA).

**Supplementary Materials:** The following are available online at http://www.mdpi.com/2073-4441/11/4/839/s1, Figure S1: WEAP Model Years and Time Steps, Table S1: Computation of Nash-Sutcliffe model efficiency coefficient (NSE).

**Author Contributions:** Conceptualization, T.M.T.; Formal analysis, T.M.T.; Investigation, T.M.T.; Methodology, T.M.T., P.M. and A.N.; Supervision, P.M. and A.N.; Writing—original draft, T.M.T.; Writing—review & editing, P.M. and A.N.

**Funding:** This research received no external funding.

**Conflicts of Interest:** The authors declare no conflicts of interest.

## Abbreviations

| | |
|---|---|
| BGR | Bundesanstalt für Geowissenschaften und Rohstoffe/Federal Institute for Geosciences and Natural Resources/Germany |
| CSO | Central Statistics Office |
| CRC | Chongwe River Catchment |
| DEM | Digital Elevation Model |
| ET | Evapotranspiration |
| GIS | Geographical Information System |
| GReSP | Groundwater Resources Management Support Programme |
| GW | Groundwater |
| LWSC | Lusaka Water and Sewerage Company |
| $m^3$/s | Cubic meters per second |
| MEWD | Ministry of Energy and Water Development |
| $Mm^3$ | Million Cubic Meters |
| RC | River Catchment |
| RH | Relative Humidity |
| SASSCAL | Southern African Science Service Centre for Climate Change and Adaptive Land Management |
| SEI | Stockholm Environment Institute |
| UNZA | University of Zambia |

| USGS | United State Department of Geological Survey |
| WARMA | Water Resources Management Authority |
| WEAP | "Water Evaluation And Planning" system |
| WS | Water Supply |
| WMO | World Meteorological Organization |
| WW | Wastewater |
| ZMD | Zambia Meteorological Department |

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
