# Peer review of "Hydrological Modelling and Water Resources Assessment of Chongwe River Catchment using WEAP Model"

_water, doi:10.3390/w11040839_

Round 1

Reviewer 1 Report

Comments:

The paper addresses water utilization issues, including source, and quantifies water budget components for a case study from Zambia. Such kind of studies are useful to enhance our understanding of hydrologic system especially in developing countries.  While the paper suits for publication, the authors need to address some of the following comments.

1.       In its present form, the paper looks like technical report. So the authors need to narrow down the manuscript, for example, by moving some tables and figures to supplemental materials. For example, the authors reported Table 1 and Figure 7 that both show the same data. They need to report either table or figure.

2.       Most of the figures are from model interface screenshot and their qualities are too low. The authors advised to improve the resolutions of their figures throughout the manuscript.

3.       There are so many typos throughout the manuscript. For example, in line 21, the authors typed “pick” to say “peak”. Such typos need to be fixed by carefully reading the manuscript.

4.       The authors sometimes used acronyms without spelling out in the first call, e.g. line 19, WEAP and GIS. This was happened in many places that need to be fixed.

5.       The authors should consider space between numbers and units, e.g., see line 26 (321.94Mm3). This was the case in many places. Please fix it throughout the manuscript.

6.       The methods section is too long. The author should consider moving some explanation of the data to supplemental materials.

7.       Add river network to figure 1 and combine with figures 2 and 3

8.       Ln 145, there is difference between streamflow and outflow, whereby the latter usually used for water harvesting structures such as reservoirs. For their study, the appropriate terminology is “streamflow”

9.       Catchment and basin was interchangeably used throughout the manuscript while there is a clear difference between the two. Please stick to one considering your study area

10.   Figure 6, plot outside the model interface and report high resolution figure. This should go to the other similar figures.

11.   Ln 247, add reference and evapotranspiration should be discussed after temperature, relative humidity, wind speed, and solar radiation data.

12.   Ln 294-296, repeated in the previous paragraph

13.   Figure 10, rename the legend and remove the title

14.   Ln 280, please spell out ERDAS and ArcGIS in the first call. This was the case for other acronyms

15.   Ln 299-303, please clarify how this information was used in the model or study.

16.   Ln 305-330, please move to introduction section

17.   Ln 354, methods and results mixed

18.   Ln 406, you already showed precipitation in methodology. Finish there and WEAP is not modeling precipitation but uses as input to partition into different components

19.   You may consider deleting Figure 15 and 16 or combining them together with Figure 17

20.   Table 7, reduce the decimal places to 1.

21.   Ln 485, round the number to one decimal place and consider to use scientific value

22.    Ln 546, explain what that value tells us

23.   Report Table 9 or Figure 26

Author Response

Dear Reviewer 1,

Top of Form

We would like to thank you for your thoughtful reviews of the manuscript. Your comments were very helpful to improve the quality of our manuscript. This report responds to your comments accordingly. The first comments from referees are given then the corresponding author’s response and changes in the manuscript (if there are changes) are followed in Red. The Line, Figure and Table numbers both for the comments and in our response refer to the original manuscript that was reviewed.

Thank you

Authors

Reviewer 2 Report

The paper “Hydrological Modelling and Water Resources Assessment of Chongwe River Catchment Using WEAP Model and GIS” by Tena et al. used WEAP model to estimate streamflow and many other water balance components in Chongwe River Catchment in Zambia. Below are my general comments and suggestions.

1.       First of all, the paper is length. Could slim it down from reducing the number of figures and reorganize the structure of the paper (also see my 3rd comment below).

2.       Second, introduction needs to bring out the significance of this study. What is a big picture here?

3.       Third, there are mixed methods and results in the paper. Need to move all methods to one session.

Some specific comments:

Figure 1 and 2 can be combined.

Figure 4 is not legible.

Figure 5 and 6 (and a few other similar ones) didn’t show the legends on the top-left corner clearly.

Figure 7, 8 and 9 can be combined into one.

Figure 10 can be combined with figure 1 and 2.

Figure 13 and 14 WEAP simulated flow are much higher than the observed flow during the high flow condition. Why is it the case? Also although r2 is good, it is not enough to assess the goodness of fit. Suggest using Nash-Sutcliffe value.

Figure 17 and Table 5 give repetitive information.

Figure 18 how does the WEAP simulated streamflow compare with the measured values at the River outlet?

Figure 23 suggest removing the numbers from the plot. If they are already included in a table, it is not necessary to have them listed again in the figure

Figure 24 is this model result? Please clarify.

Author Response

Dear Reviewer 2,

Top of Form

We would like to thank you for your thoughtful reviews of the manuscript. Your comments were very helpful to improve the quality of our manuscript. We have attached our response in pdf.The report responds to your comments accordingly. The first comments from referees are given then the corresponding author’s response and changes in the manuscript (if there are changes) are followed in Red. The Line, Figure and Table numbers both for the comments and in our response refer to the original manuscript that was reviewed.

Thank you

Authors 

Reviewer 3 Report

This manuscript presents an analysis of the water balance of the Chongwe River in Zambia using the WEAP model.  The analysis of long term data sets and estimations of water use and abstraction provide information to make recommendations on the management of the Chongwe watershed.  The topic was interesting and is important work.  However, the manuscript needs substantial improvement in organization to help readers better understand the process leading to the results and recommendations. 

I have organized my suggestions by section

Major changes required:

There are too many figures, many or unnecessary.  Most of the figures are very poor graphics or screen shots from the model that need a better presentation.

Data and Methods needs reorganizing to be more logical toward describing the WEAP modelling process for Chongwe River.  My recommendations are to start section with a description of WEAP, water budget, background; moving 2.2, 2.2.1, 2.2.2 to start of section.  Then describe the inputs.  Consolidate all of the sub-sections on inputs into 2 sections, 1) climate and physical, 2) hydrologic. When describing components of the water cycle for WEAP analysis please connect the inputs with the reason or description they are used in the modelling.  For example, you provide a lengthy description of land-use and cover but do not explain how it is used in the model.

Many of the results (section 3) have methodological descriptions, these should be moved into section 2

Some recommendations at end are not well supported by text in section 3.  Please put more effort to bring forward model results that allow the reader to understand the recommendation.  For example recommendation on forest land retention, yet no results on change of forest cover type shown. 

Citations not in Water format; should be numbered in text in order of appearance.

Here are a few specific edits by section

Abstract

Lines 10 and 11: end sentence after Zambezi Basin; delete remainder of sentence.

Line 15:  delete “ one of the challenge of the entire community” replace with “is the main water resource challenge.”

Line 16 delete “entire”

Line 17 delete “thus”

Line 19 provide full model name with abbreviation in parenthesis

Line 22 replace “pick” with “peak”;  delete the word “for” ; after the word calibration add “of the model.”

Line 23 delete “purposes.”

Line 24-32 put the word “estimated” before “mean annual”

Line 29-30 delete “ result of the model”

Line 30 end sentence after 10.32 m3/s.  Capitalize “The” to start new sentence.  Add “Chongwe River” after “The” in new sentence.

Add a sentence before last sentence of abstract briefly stating research objective

Introduction

Line 74-76  Move sentence the starts “Effective and responsible…” to the start of the next paragraph starting on line 77.

Line 77  delete “Therefore,”

Section 2:

Remove Figures 3, 6, 7, 8, 9 put summarization in tables or brief summarization in text.  For ET calculations? 

Many words are improperly capitalized.  Precipiation, Evapotranspiration, etc. are not proper nouns and do not get capitalized. 

Section 3:

Section 3.1 is unnecessary and is not a result, please delete.  Water balance already described succinctly in WEAP description. 

Many of the result section have methodological descriptions, these should be moved to section 2.  For example, in section 3.2.1 lines 333-338, line 346 are methods. 

Line 346 is repeating info presented in section 2. 

The entire section 3.2.3 and 3.4 are methodology. 

Lines 391-394 move to section 2. 

Figure 15, 16, 18, 22, 23, 25 appear to be model screen shots.  These figures need a more appropriate presentation.  Text needs to be larger and readable.  Legend scaled to readable size.  Remove software toggle buttons, etc.

Table 5 appears to have problems with incorrect significant figures.  You do not have results accurate to 1/100000 mm.  The captions does not indicate it this is WEAP output, I believe it is. 

Figure 8 appears to be model screen shots.  These figures need a more appropriate presentation.  Text needs to be larger, to be readable. 

Delete figure 19, only provide table of monthly results.

Table 6 and 7 has significant figures incorrect.  3 decimal places at most, but 2 might be better.

Delete Figure 21 already described in the text. 

Tables 7 and 8 values need comma for each order of magnitude.  E.g. change 123456789 to 123,456,789. Repeat for similar large numbers in text.

Figure 24 poor quality graphic

Section 3.7.1 delete, it is unnecessary. 

Figure 26 is a very good graphic!

Section 4

Move lines 578-584 out of the conclusion, put at end of section 3 as a subsection.

Author Response

Dear Reviewer 3,

We would like to thank you for your thoughtful reviews of the manuscript. Your comments were very helpful to improve the quality of our manuscript. We have attached our response in pdf. The report responds to your comments accordingly. The first comments from referees are given then the corresponding author’s response and changes in the manuscript (if there are changes) are followed in Red. The Line, Figure and Table numbers both for the comments and in our response refer to the original manuscript that was reviewed.

Thank you

Authors 

Reviewer 4 Report

This paper is mainly about hydrological modeling of a basin in Zambia. Although the study area is generally considered a wet watershed but it suffers from water resources shortage in recent years and the authors tried to investigate this situation scientifically. However, the paper is not very well-written and well organized. It needs some major changes before final publication:

The abstract part should be more informative while you should not include all the details. Try to show the main finding of your paper in this section.

Go through the paper and try to correct English errors. Do not use long sentences and try to use appropriate tenses for verbs/sentences.

You need to provide more detail about WEAP model and compare it with more well-known hydraulic and hydrologic models.

When using abbreviations for the first time, you have to provide the full format. For example, when using WEAP for the first time, you need to provide the full format (if any).

You need to provide your results in more scientific ways. You need to discuss your findings in a better way.

Try to compare your results with other studies and show what is your novelty.

You need to provide numbers in a way that makes sense. For example, in table 5, you have provide numbers and it does not provide a good and informative data to the readers. You need to provide better captions for your graphs and tables.

Your graphs are pretty simple and not very informative.

Your conclusion should provide more detailed data about your findings. You can put most of the conclusion part in introduction section.

The paper needs major revision

Author Response

Dear Reviewer 4,

We would like to thank you for your thoughtful reviews of the manuscript. Your comments were very helpful to improve the quality of our manuscript. We have attached our response in pdf.  The report responds to your comments accordingly. The first comments from referees are given then the corresponding author’s response and changes in the manuscript (if there are changes) are followed in Red. The Line, Figure and Table numbers both for the comments and in our response refer to the original manuscript that was reviewed.

Thank you

Authors 

Round 2

Reviewer 2 Report

The quality of the revised paper “Hydrological Modelling and Water Resources Assessment of Chongwe River Catchment Using WEAP Model and GIS” by Tena et al. has improved significantly. I am happy to recommend this paper for publication after the following points are addressed and/or considered.

1)      I would suggest removing GIS from the title.

2)      Line 159-160, define delta S.

3)      Line 268, need to add how these three equations were derived. Are they statistically significant? Did you do a trend analysis? Using the bar diagram with the fitted linear relationships is quite confusing. What is x in the equation? Same issue is also applied to Figure 7.

4)      Section 3.1.5 model performance and Figure 11, please clarify observed monthly mean streamflow was from which year to which year. Is it possible to show time series data? What time step is the WEAP flow output? Maybe do the monthly mean for each year when observed data are available.

5)      Table 9 is not necessary. Would suggest deleting it.

6)      Line 381, there is no Figure 27.

7)      Line 420, should read 4.2. Recommendation

Author Response

Dear Reviewer 2 We would like to thank you for your constructive and thoughtful reviews of the manuscript. Your respective comments are instrumental and helpful to improve the quality of our manuscript. Thank you Authors

Reviewer 3 Report

This manuscript presents an analysis of the water balance of the Chongwe River in Zambia using the WEAP model.  The analysis of long term data sets and estimations of water use and abstraction provide information to make recommendations on the management of the Chongwe River catchment.  The topic was interesting and is important work. 

I am very impressed with how much improved the article is since my first review.  I feel this article is worthy of publication provided minor changes are made.  Below are my main recommendations.

1)      One thing that will greatly improve this article is to provide a summarization of the water budget components over time.  You provide temporal change for individual components of the water budget, e.g. precipitation, evapotranspiration, abstraction, etc.  You also provide a nice figure on the 2017 budget.  Your conclusions and recommendations would be better supported by adding a table showing the quantification of the different elements in equation 1.  I am guessing you would need an additional row for the error in closing the budget.  This could be done with as few as 2 time periods, 1984 and 2017.  By doing this your discussions would be supported within the context of the entire water budget, not just changes in individual elements.

2)     For actual evapotranspiration, potential evapotranspiration, and precipitation the results state a percentage change over time e.g lines 272, 273, 283.  Based on the equations and data shown in Figure 6 and 7 the changes are in units of Mm3 not percent.  Please change all circumstances and discussion that state these differences as percent.

3)     Please state all results in the past tense.  Your study ended in 2017, that is in the past.  E.g use “was” instead of “is” etc.

The following are editorial recommendations:

Citations and references should be numbered in order of appearance not alphabetically.  See author instructions for Water. https://www.mdpi.com/journal/water/instructions

Key words:  delete “catchment”, this is not informative.  Consider adding “Zambesi River”.

Line 49 replace “very few portions” with ”little”

Line 64 temperatures (plural)

Line 65-66 delete “ for various uses”

Line 70 delete “available”

Line 107 delete “few”

Line 112  rewrite as “ productive aquifers,  Class D are moderately productive aquifers, Class E are minor aquifers with”

Line 113 end sentence at “resources.”  Start new sentence with “These aquifers cover the larger part…..”

Figure 2 remove the title at top it is redundant with the caption

Figure 2 remove “_” between Chongwe and Aquifers in legend.

Line 128 delete “Which can”  replace with “WEAP can”

Line 138 replace “seats” with “sets”

Line 139 replace “in organized” with “in an organized”

Line 140 put a comma after “side” then delete “and on an equal footing with”

Delete lines 151 -158 to end of sentence “and area”.  These are not necessary, this is general knowledge and unnecessary.

Combine Lines 158-162 into 1 paragraph.

Line 173 insert “from Chongwe River Catchment” after “inputs” then move lines 168-173 starting with equation 1 before paragraph on line 163.

Line 176 insert “air” before “temperature”; please do this throughout the manuscript to clarify what temperature is measured.

Line 177  the cloudiness fraction associated with solar radiation should be explained.  Is solar radiation only quanitified by a cloudiness fraction?  This is how it reads.

Line 181 insert “air” before “temperature”; please do this throughout the manuscript to clarify what temperature is measured.  This will be the last time I mention this.

Table 2, please organize so it does not break over 2 pages.

Line 190 move “Imagine” into the parenthesis after “ERDAS”  i.e. (ERDAS Imagine)

Line 190 the term “and Arcmap 10.3.” does not fit in this context.  The action of the sentence is “land use was generated from”, was data from Arcmap generated or analyzed in Arcmap?  If Arcmap was only a tool and not a source of data please re-word.

Figure 3 please make the legend font larger so it is easier to read.  Please delete “Land Use_ Land Cover Cover…..” in legend, it is already in caption.

Lines 198-201 move to Line 195 and before Figure 3.

Line 219 provide a citation for the source of the software, do this for Arcmap as well earlier in paper, typically (Company, City, Country) where software originates.

Line 230  please clarify what the 77.4% is from, is it 77.4% of Ngwerere flow, of Chongwe flow?

Line 235  replace “ using a set of layers of” with “from a”

Figure 5 needs a legend, scale, and north arrow

Line 257 change Qs and Qo to Qst and Qot respectively

Line 257-258: reword after the comma  to:  “Qst and Qot are simulated and observed streamflow at time t respectively.”

Line 268 change “releases” to “was estimated to have”

Line 272 change “is” to “has been” ; use past tense when describing results from data in the past.  Change 0.24% to 0.24 Mm3

Figure 6 Change the color of the trendline for precipitation to the color blue to match the bar graph color for precipitation, AET and PET symbols are in this color scheme.

Line 281 change “is” to “was”  this is the last time I mention change in tense.  Please fix all instances.

Line 282-283  change the sentence to:  “ The average annual streamflow at the outlet was 321.94Mm3/year and has increased at a rate of 0.13 m3/s per annum (Figure 7).”

Line 302, 304 and tables 6 and 7.  Consider changing these numbers to Mm3  you are willing to use that unit for other budget variables.

Line 345 change % to Mm3 ; insert author(s) name before citation number [3].  E.g. “as stated by Smith et. al [1]”.

Line 378 change % to Mm3

Figure 10:  add “for 2017” at the end of the caption.

Line 328 insert tables of water balance variables from equation 1 over time. See main recommendation 1 at start of this review.

Line 383 delete “also”

Line 384 change “as” to “an”

Line 388 the phrase 2.6% change in storage should be supported.  Again adding a water balance table to manuscript will help these discussions.

Lines 398-399 delete “an adequate amount of”

Line 404 There needs to be a discussion of changes in water balance by upper, middle, or lower portions of catchment in the main body of the manuscript to make this conclusion.

Line 418 delete “plant trees and” insert “and increase” after “protect”

Author Response

We would like to thank you for all of your constructive and thoughtful reviews of the manuscript. Your respective comments were instrumental and helpful to improve the quality of our manuscript.

Thank you

Authors

Reviewer 4 Report

It can be published after editor final decision.

Author Response

We would like to thank you for your thoughtful reviews of the manuscript. Your respective comments were helpful to improve the quality of our manuscript.

Thank you

Author
